# Comparative Evaluation of the Autonomic Response to Cognitive and Sensory Stimulations through Wearable Sensors

**DOI:** 10.3390/s19214661

**Published:** 2019-10-27

**Authors:** Alessandro Tonacci, Lucia Billeci, Elisa Burrai, Francesco Sansone, Raffaele Conte

**Affiliations:** 1Institute of Clinical Physiology—National Research Council of Italy (IFC-CNR), Via Moruzzi 1, 56,124 Pisa, Italy; lucia.billeci@ifc.cnr.it (L.B.); francesco.sansone@ifc.cnr.it (F.S.); raffaele.conte@ifc.cnr.it (R.C.); 2School of Engineering, University of Pisa, Largo Lucio Lazzarino 1, 56,122 Pisa, Italy; elisa.b91@live.it

**Keywords:** autonomic nervous system, ECG, olfaction, skin response, stress

## Abstract

Psychological stress is known to activate the autonomic nervous system (ANS), thus representing a useful target to be monitored to understand the physiological, unconscious effect of stress on the human body. However, little is known about how differently the ANS responds to cognitive and sensory stimulations in healthy subjects. To this extent, we enrolled 23 subjects and administered a stress protocol consisting of the administration of sensory (olfactory) and cognitive (mathematical) stressors. Autonomic parameters were unobtrusively monitored through wearable sensors for capturing electrocardiogram and skin conductance signals. The results obtained demonstrated an increase of the heart rate during both stress protocols, with a similar decrease of the heart rate variability. Cognitive stress test appears to affect the autonomic parameters to a greater extent, confirming its effects on the human body. However, olfactory stimulation could be useful to study stress in specific experimental settings when the administration of complex cognitive testing is not feasible.

## 1. Introduction

All living beings, including humans, regularly respond to a plethora of stressful stimuli, named stressors, that continuously challenge their homeostasis, which is the complex and dynamic equilibrium that makes life functions possible [1].

To properly respond to the onset of stressors, a quite complex system, named the stress system, has been developed, providing appropriate neuroendocrine responses both at central and peripheral levels [1]. As such, the stress system has both central and peripheral components, interacting with a number of other systems at various levels. Central components include the parvocellular neurons, the neurons of the paraventricular nuclei of the hypothalamus, the corticotropin-releasing hormone neurons, and the neural groups at medullar and pons levels. On the other hand, peripheral components include the neuroendocrine hypothalamic–pituitary–adrenal axis, the efferent systemic sympathetic–adrenomedullary systems, and other components functioning under the control of the parasympathetic system [2].

Therefore, it is quite understandable that both sympathetic (SNS) and parasympathetic nervous systems (PNS), representing the two functional components of the autonomic nervous system (ANS) and influencing the activity of several physiologic systems (cardiovascular, respiratory, renal, gastrointestinal, neuroendocrine, etc.), are well involved in the stress response. In particular, the SNS is known to play a key role in the so-called “fight or flight” response, through the adrenal medulla, secreting epinephrine and norepinephrine. Conversely, the PNS increases or contrasts the activity of the SNS if needed [2]. Both the SNS and PNS transmit neural signals by acetylcholine and norepinephrine neurotransmitters, as well as by several neuropeptides, adenosine triphosphate, or nitric oxide [3].

A wide range of autonomic effects has been noticed based on the type of stimuli presented. Notably, sensory stimulation, particularly olfaction [4], was reported to cause a significant autonomic response, especially when the stimulations are based on particularly unpleasant odors. Meanwhile, even at a lesser extent, also common odors can elicit autonomic variations in healthy individuals [5].

A recent review has also outlined the autonomic involvement in a wide range of cognitive tasks, demonstrating the deep influence of the ANS in cognitive functioning, albeit not drawing exhaustive conclusions concerning cognitive subdomains because of the relatively small number of studies published to date [6]. However, to the best of our knowledge, a comparative analysis between the ANS response to sensory and cognitive stimuli, possibly using the same protocol and instrumentation, is lacking in cognitively intact subjects.

Fortunately, thanks to the quick technological advancements in information and communication technologies, notably those applied to the field of healthcare, several solutions are nowadays available to unobtrusively extract key information related to the ANS functioning. Among them, wearable sensors are massively used also in naturalistic settings, thanks to their excellent portability, low weight and (relatively) low cost [7,8], actually representing a useful alternative to this extent.

Understanding the usefulness of wearables to capture even subtle information related to the physiological processes occurring in the human body could be critical to explore new frontiers of clinical and translational research, enabling such minimally invasive, user-friendly systems to be employed for self-monitoring purposes within a framework of patient empowerment and personalized medicine.

Among the information possibly retrieved using wearables, physiological signals, including an electrocardiogram (ECG) and galvanic skin response (GSR), could be greatly helpful in characterizing the ANS and its behavior, as already demonstrated [9].

Under those premises, the present work aims to investigate the autonomic variations possibly occurring during cognitive and sensory (olfactory) tasks, by proposing a minimally invasive approach to study this specific function, drawing potentially useful conclusions for practical applications dealing with the treatment of stress and ANS hyper-reactivity.

In this paper, the materials and methods employed are defined in Section 2, Section 3 briefly outlines the numerical results obtained, the discussion is in Section 4, and Section 5 contains all the concluding remarks.

## 2. Materials and Methods

### 2.1. Study Population

This study involved 23 cognitively intact subjects (15 males, 8 females, aged 19–49 years old). A power analysis was computed with SPSS Statistics v.23 (IBM Corporation, Armonk, NY, USA) for the whole statistical analysis. Power analysis calculations were performed using data from the Heart Rate (HR)/ Heart Rate Variability (HRV) and GSR parameters. Among them, we used the most conservative of all those parameters, which was the HR. Similarly, calculation of the Glass’ delta was done to evaluate the effect size, taking into account the HR as well, and was performed instead of using Cohen’s d since the distributions had different standard deviations. This latter analysis was conducted by the online software Social Science Statistics (available at: https://www.socscistatistics.com/effectsize/default3.aspx).

All the subjects gave their informed consent for inclusion before they participated in the study. The study was conducted in accordance with the Declaration of Helsinki (current version available at: https://web.archive.org/web/20091015082020/http://www.wma.net/en/30publications/10policies/b3/index.html).

Inclusion criteria were a minimum education level of 13 years (secondary school completed). Exclusion criteria included the presence of associated neurological, psychiatric, cardiovascular, respiratory, metabolic and orthopedic conditions, sensory (visual or olfactory) impairment reported, pharmacological treatments (with cardiovascular and/or respiratory drugs, anxiolytics, antidepressants), or inability/unwillingness to sign the informed consent.

All the subjects were asked to undergo physiological signal measurements at rest and during the administration of an olfactory battery, as well as during a cognitive task, and to compile a quick questionnaire related to the odors sniffed.

### 2.2. Signal Acquisition

For this study, participants were equipped with wearable devices for the acquisition of physiological signals, including electrocardiogram (ECG) and galvanic skin response (GSR).

Both signals were captured with sensors manufactured by Shimmer Sensing, Inc. (Dublin, Ireland), the ECG signal at 500 Hz by means of the single-lead Shimmer™ ECG Unit and the GSR signal at 51.2 Hz with the Shimmer3™ GSR+ Unit, according to a protocol previously published [10].

The devices were connected by Bluetooth to a laptop running a proprietary graphical user interface enabling signal acquisition management.

Both the ECG and GSR signals were recorded during five different phases, as detailed below.

(i)Baseline (3 min): basal measurement. The subject, seated in a comfortable chair, was asked to relax during this phase;(ii)Task I (4 min): olfactory task I. The subject, comfortably seated, was asked to sniff a battery of six odors (as detailed in the following section);(iii)Task II (5–10 min, depending on the subject): cognitive task. The subject, comfortably seated, was asked to complete a cognitive task administered through a user interface on the laptop (later detailed);(iv)Task III (4 min): olfactory task II. Same procedure as Task I. The olfactory test was performed before and after the cognitive task in order to assess whether the prior administration of a cognitive task would affect the ANS response to the olfactory task;(v)Recovery (3 min): post-task basal measurement. The subject was asked to relax during this phase, similarly to the baseline.

### 2.3. Olfactory Stimulation and Related Questionnaires

As reported, Task I and Task III were related to the presentation of odorous compounds to be investigated in terms of the ANS activation in the volunteers enrolled for the present study.

In order to obtain reliable data, we performed this task proposing six odors from the well-grounded Sniffin’ Sticks Test [11]. More specifically, the odors proposed were pineapple, butanol, mushrooms, peach, coconut, and fish, representing a somewhat balanced mix of pleasant and unpleasant substances for a non-anosmic group of subjects (see, for example [12]).

During the protocol, each odor was administered using felt tip pens for 10 s, with a 30 s delay between two consecutive stimuli, in order to let the nostrils of the volunteer to be properly purged from the previous compound (see [13]).

During the 30 s resting period, the volunteer was asked to provide a score concerning the perceived pleasantness of the odor sniffed on a 9-item Likert scale, ranging from “−4” (most unpleasant) to “+4” (most pleasant), with the “0” as the neutral descriptor.

### 2.4. Cognitive Stressor Administration

As for the cognitive task (Task II), the validated, largely used Trier Social Stress Test (TSST) [14] was administered through a developed script within the Inquisit 5 software (Millisecond Software, LLC, Seattle, WA, USA). TSST, divided into three trials, requires the volunteer to quickly solve simple mathematical operations, providing as output data the total number of correct answers, the total time elapsed, and the percentage of correct answers.

### 2.5. Signal Analysis

#### 2.5.1. ECG

ECG signal was analyzed through an interface developed in Matlab (Mathworks, Natick, MA, USA), extracting the tachogram from the raw ECG signal according to the Pan–Tompkins algorithm [15] and, after artifact removal, allowing the operator to calculate the time- and frequency-domain features associated with the signal, as demonstrated in [8,9,16].

The features extracted are displayed in Table 1:

All the frequency domain features were extracted using the power spectral density estimation according to the Welch method [19].

#### 2.5.2. GSR

GSR signal was analyzed using Ledalab V3.4.9 (General Public License (GNU)), a Matlab-based tool enabling the user to extract typical features of the GSR signal, notably its tonic and phasic components [20]. Specifically, though both the above-mentioned components were extracted, for this work only the tonic phase was analyzed, since the rationale of the study aimed at comparing the signal in the different phases rather than evaluating the response to a given single stimulation. The information related to the GSR signal can be used for the assessment of SNS activity (see, for example, [21]).

### 2.6. Statistical Analysis

First, a Shapiro–Wilk Test was conducted to check the normality of distributions for each parameter analyzed. In case data were found to deviate from the normality, parametrical tests (repeated-measures ANOVA and, in case of significance, Student’s *t*-test to evaluate significant differences in pairwise comparisons) were performed. Otherwise, non-parametrical approaches (Friedman’s test and, in case of significance, Wilcoxon signed-rank test) were employed to compare the different phases and the single phases one-by-one, respectively.

Concerning correlations between autonomic and olfactory (or cognitive) parameters, in case of normal distributions, Pearson’s test was used, otherwise Spearman’s test was preferred. Post hoc analysis was carried out using the false discovery rate (FDR) test. Statistical significance was considered for *p* < 0.05.

## 3. Results

Power analysis revealed a power level of 0.95 (F = 7.477, α = 0.05, power = 0.95). At the same time, the calculated effect size, according to the Glass’ delta, resulted to be medium (g = 0.51).

Data about the autonomic parameters above mentioned, with *p*-values reported in the comparison across phases and comparing the single phases one-by-one, are reported in Table 2.

All the trends of significant ECG data are displayed in Figure 1.

The GSR tonic trend is otherwise displayed in Figure 2.

The correlation analysis failed to reveal significant relationships between the autonomic parameters and odor pleasantness scores, whereas pNN50 during Task III was seen to be positively correlated, even after FDR correction, to the number of trials completed in the TSST (r = 0.607, *p* = 0.002) (Figure 3).

## 4. Discussion

Our study, conducted on a limited, though well characterized, number of young, healthy volunteers, aimed at comparing the autonomic response to cognitive and sensory tests to the best of our knowledge, and tried to fill in a significant gap in the existing literature.

With respect to the baseline, the presentation of the six different olfactory stimuli was seen to elicit a significant response in terms of activation of the sympathetic branch of the ANS. Having reported that the olfactory compounds used included odors normally judged as both pleasant and unpleasant [12], our findings are somewhat similar to the retrievals of Bensafi et al. [22], limited to unpleasant odors. In addition, we also failed to find significant correlations between autonomic parameters and odor pleasantness, confirming that pleasant versus unpleasant odors do not cause significantly different autonomic responses in healthy subjects [22].

After the olfactory stimulation, the TSST further increased the SNS activity, at the same time highlighting a somewhat parasympathetic withdrawal, as demonstrated by the simultaneous decrease of both SDNN and pNN50. Despite not being extensively studied with respect to the TSST, this kind of ANS behavior is consistent with the results by Boesch et al. [23], which, with a slightly different protocol, found an increase in HR together with a contextual decrease of HRV features during the administration of the TSST for groups (TSST-G). Similar results in terms of HR augmentation and HRV reduction under TSST stress were also found in several other protocols (see [24,25]), strengthening the scientific value of our findings.

The comparison between ANS responses to olfactory stimulation and to the TSST protocol made it possible to compare Task I and Task II, as well as Task II and Task III responses, and highlighted a much higher contribution to the SNS activation borne by the TSST. Therefore, TSST was confirmed to represent a powerful psychosocial stress task capable to induce an acute stress response of all main physiologic stress systems [23].

In addition, olfactory stimulation also elicited ANS activation (particularly SNS) The magnitude of this response was significantly smaller than that occurring with TSST, highlighting its usefulness in stress monitoring of subjects unable to undergo complex mathematical stress tests, limited to some particularly controlled experimental settings.

Interestingly, the trends noticed in the phase transition between Task III and recovery suggest an overall ANS withdrawal after a massive psycho-sensory stress protocol, being that both SNS and PNS activity decreased at this stage.

Finally, the only significant correlation revealed that subjects who more quickly completing the TSST protocol displayed higher PNS activation in the following phase. This result could be speculated in terms of sustained hyper-reactivity of the PNS, as a marker of increased selective attention and higher-order behavior and cognition (e.g., see [26]).

### Limitations

All these findings should be taken into account in light of some limitations.

First, the relatively small sample size does not allow drawing considerations based on subgroups (e.g., gender- or age-based, etc.). This fact did not allow investigating the possible effects of confounding factors, including gender-specific ones (i.e., the sexual-hormone influences for female subjects, menstrual cycle phases, etc.) or variables related to some physical characteristics of the subjects (i.e., the body mass index), even though none of the volunteers was apparently under- or overweight. Second, due to the necessity of keeping the overall length of the protocol reasonably low, we did not consider long-term ANS measurements that could be more precise in characterizing the overall stress state of the volunteers. Third, for the same reason, we decided not to administer stress-related questionnaires, such as Visual Analogue Scale (VAS) or State-Trait Anxiety Inventory (STAI), to monitor both state and trait anxiety. Such protocols can be applied in future studies, allowing assessing possible correlations between the subjective, perceived stress and the variations of the ANS parameters.

Fourth, for the reason mentioned above, we decided to keep the recovery phase reasonably short. With a longer duration of this phase, one could have inferred the duration of the autonomic effects of both cognitive and sensory stimulation. This limitation could be taken into account and solved in future studies.

Fifth, the somewhat counterintuitive results retrieved in studying time- versus frequency-domain features were partially explained by the contrasting literature explanation regarding the physiological processes underlying frequency-domain variations, as explained in the signal analysis subsection of the Materials and Methods. Further studies conducted on larger samples, and using other time-domains and more robust ECG features, including RMSSD and the Cardiac Vagal Index calculated from the Lorenz plot [18], could also help in clarifying this point.

Finally, the administration of both pleasant and unpleasant odors would have represented a possible confounding factor if one’s aim was to control the selective ANS response to one of the two categories. In light of such considerations, future protocols could be structured starting from the selection of odors based on their hedonic tone, even referring to pleasantness data reported in previously published studies.

## 5. Conclusions

Despite the limitations mentioned above, the present study demonstrated the reliability of using wearable sensors for unobtrusively monitoring the ANS activity during a stress protocol. Furthermore, the sensitivity of this methodology also allowed detecting slight autonomic changes like the ones occurring when administering a mild sensory stimulation.

With this approach, the mathematical TSST stress test was seen to be a powerful autonomic activator, potentially usable in all protocols requiring a strong stressor to characterize the physiological, unconscious response, for example, in subjects with anxiety disorders or in workers subjected to highly stressed tasks. In particular conditions, when the subject is not capable of carrying out complex cognitive tasks (e.g., in patients with dementia or with important cognitive impairments), but displaying an intact sensory path, the administration of an olfactory stimulation could represent a useful alternative to complement this information.

Future studies taking advantage of this approach can be performed either on healthy or diseased subjects, increasing the study population and tailoring the selection of odorous compounds to be used in order to evaluate specific response patterns to given compounds. The evaluation of other biosignals, including electroencephalogram (EEG) and electromyography (EMG), is also desirable.

## Figures and Tables

**Figure 1 sensors-19-04661-f001:**
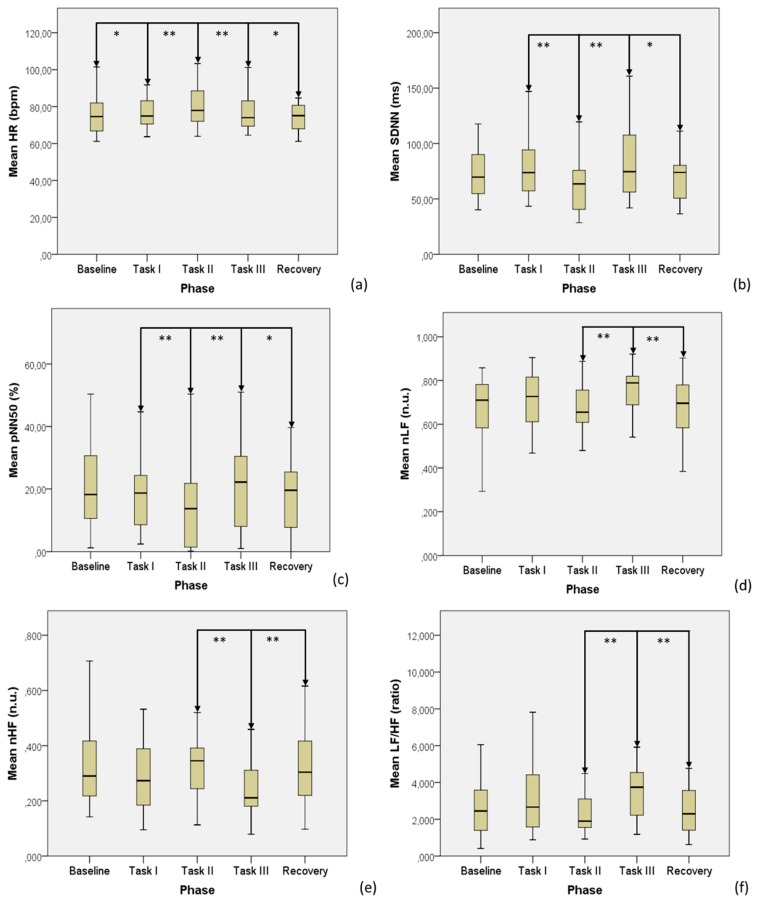
Significant ECG data across the test phases: (**a**) HR, (**b**) SDNN, (**c**) pNN50, (**d**) nLF, (**e**) nHF, (**f**) LF/HF. *: *p* < 0.05, **: *p* < 0.01.

**Figure 2 sensors-19-04661-f002:**
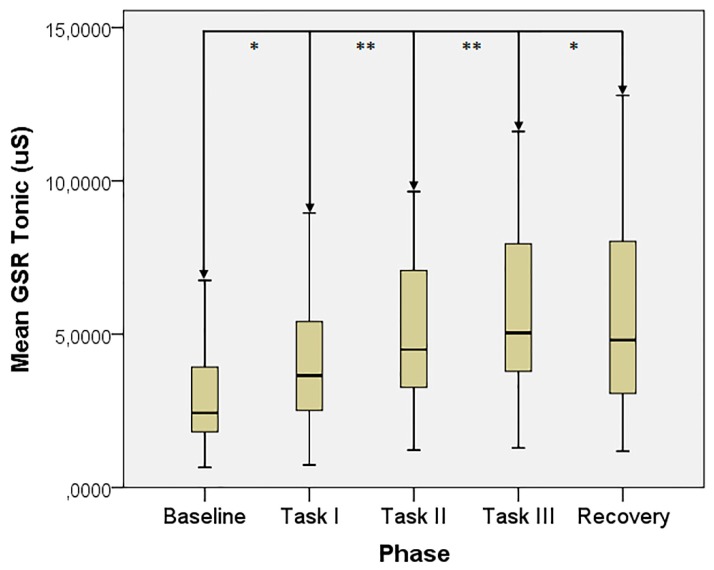
GSR tonic value across the test phases. *: *p* < 0.05, **: *p* < 0.01.

**Figure 3 sensors-19-04661-f003:**
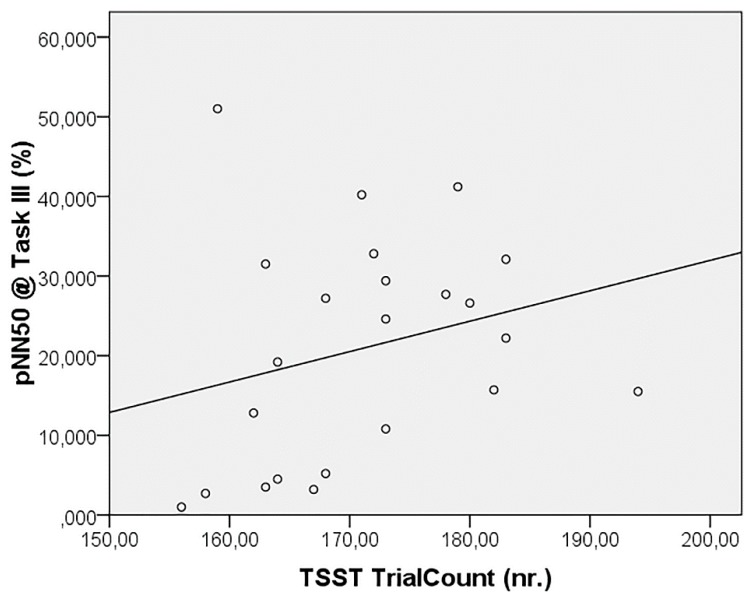
Correlation between TSST trial count and pNN50 during the task.

**Table 1 sensors-19-04661-t001:** Time- and frequency-domain features extracted from the ECG signal.

**Time Domain**		
**Feature**	**Acronym**	**Physiological Significance**
Heart rate	HR	Number of heart pounds per time unit, expressed in bpm. The HR is normally related to the SNS activation of the ANS
Standard deviation of normal-to-normal R–R intervals	SDNN	Measure of heart rate variability (HRV), expressed in ms. The SDNN is normally affected by both SNS and PNS components, but largely depends on testing conditions, with the PNS activity being predominant during short-term resting recordings [17]
Changes in successive normal sinus (NN) intervals exceeding 50 ms	pNN50	Measure of HRV, expressed as a percentage. Overall, pNN50 is closely correlated to the PNS activity [17]
Cardiac sympathetic index	CSI	Measure of SNS activity derived from the Lorenz plot analysis [18]
**Frequency Domain**		
**Feature**	**Acronym**	**Physiological Significance**
Normalized component of the power spectral density of the ECG signal at low frequency (0.04–0.15 Hz)	nLF	nLF is related to both SNS and PNS activity, though, in resting conditions, the LF band reflects baroreflex activity (see [17] and related works)
Normalized component of the power spectral density of the ECG spectrum at high frequency (0.15–0.4 Hz)	nHF	nHF is related to PNS activity, though its direct relationship with the vagal tone is not always evident
Low-to-high frequency component of the power spectral density of the ECG spectrum	LF/HF	To a first approximation, LF/HF ratio is considered a marker of the balance between SNS and PNS activity, despite this assumption is not necessarily true, especially during short, resting recordings, where, as stated, the LF often reflects baroreflex activity

**Table 2 sensors-19-04661-t002:** Autonomic parameters in the different phases indicated as mean ± standard deviation (B: Baseline, T I: Task I, T II: Task II, T III: Task III, R: Recovery; N/A: Post hoc comparison not performed due to nonsignificance of the Friedman’s test; *: *p* < 0.05, **: *p* < 0.01).

Feature	Baseline	Task I	Task II	Task III	Recovery	*p*-Value (Friedman’s Test)	*p*-Value B vs. T I	*p*-Value T I vs. T II	*p*-Value T II vs. T III	*p*-Value T III vs. R
HR, bpm	75.7 ± 10.6	77.1 ± 9.4	81.1 ± 11.8	76.3 ± 8.7	74.9 ± 9.4	<0.001 **	0.040 *	0.003 **	<0.001 **	0.014 *
SDNN, ms	74.1 ± 28.4	79.0 ± 28.0	62.5 ± 28.3	82.8 ± 31.7	74.2 ± 31.4	<0.001 **	0.094	<0.001 **	<0.001 **	0.016 *
pNN50, %	22.3 ± 16.6	19.8 ± 14.4	14.4 ± 13.0	20.9 ± 14.1	18.4 ± 14.3	<0.001 **	0.077	0.001 **	<0.001 **	0.021 *
CSI, ratio	2.95 ± 0.72	3.19 ± 0.73	3.29 ± 1.18	3.11 ± 0.66	3.18 ± 0.70	0.352	N/A	N/A	N/A	N/A
nLF, n.u.	0.67 ± 0.16	0.71 ± 0.13	0.67 ± 0.11	0.75 ± 0.10	0.68 ± 0.14	<0.001 **	0.465	0.117	0.001 **	0.005 **
nHF, n.u.	0.33 ± 0.16	0.29 ± 0.13	0.33 ± 0.11	0.25 ± 0.10	0.32 ± 0.14	<0.001 **	0.465	0.117	0.001 **	0.005 **
LF/HF, ratio	2.69 ± 1.54	3.26 ± 2.20	2.61 ± 1.84	3.82 ± 2.41	3.02 ± 2.46	<0.001 **	0.563	0.068	0.001 **	0.012 *
GSR Tonic, µS	3.46 ± 2.90	4.92 ± 4.93	5.93 ± 4.68	6.73 ± 5.18	6.53 ± 5.36	<0.001 **	<0.001 **	<0.001 **	0.001 **	0.031 *

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
