# Peer review of "Comparative Evaluation of the Autonomic Response to Cognitive and Sensory Stimulations through Wearable Sensors"

_sensors, 2019, doi:10.3390/s19214661_

Round 1

Reviewer 1 Report

The paper presents a study on 23 subjects following a stress protocol consisting of the
administration of a sensory (olfactory) and cognitive (mathematical) stressor. Autonomic parameters were unobtrusively monitored through wearable sensors for capturing electrocardiogram and skin conductance signals.

The paper is well written and well structured. Prior work is well cited. The methodology followed and data analysis used aresound. 

A few points on the protocol however:

young subjects (15 males, 8 females, aged 19-49 years: I wouldn't say 30-year old or more are young subjects! in order to evaluate the impact of the tests' order, would it have given different results if only T1 and T2 had been done in a day and then on some other day T2 and T3 would have been done? the olfactive tasks are quite long (6x10s), while the cognitive task is quite short (10mn): it would have been interesting to test various durations for both tests why after the recovery phase, the pNN and CSI do not get back to the baseline? should the recovery phase have been longer?

Editing:

Table 1 is hard to read because the newlines are at wrong places (ie, after periods or before commas)

non-diseases individuals -> healthy individuals

non-diseased volunteers -> healthy volunteers

Author Response

We would like to thank the reviewer for his/her precious comments. The comments are reported below (in plain text) with our answers in italics.

The paper presents a study on 23 subjects following a stress protocol consisting of the administration of a sensory (olfactory) and cognitive (mathematical) stressor. Autonomic parameters were unobtrusively monitored through wearable sensors for capturing electrocardiogram and skin conductance signals.

The paper is well written and well structured. Prior work is well cited. The methodology followed and data analysis used are sound.

Thank You very much for Your kind words, much appreciated.

A few points on the protocol however:

young subjects (15 males, 8 females, aged 19-49 years: I wouldn't say 30-year old or more are young subjects!

Thank You. We corrected accordingly.

in order to evaluate the impact of the tests' order, would it have given different results if only T1 and T2 had been done in a day and then on some other day T2 and T3 would have been done?

Thank You for Your comment. The nature of our study took its strength in its unobtrusiveness, both in terms of methodology used and in terms of materials (i.e., sensors) employed. Therefore, the structure of the protocol aimed at keeping the length and obtrusiveness of the protocol reasonably limited, in order to not bring too much annoyance to the patients. Therefore, we decided to keep the whole protocol within a single day of testing. In addition, by keeping the tests in separate days would require a new baseline to be recorded and to be referred to, increasing again the length of the test.

the olfactive tasks are quite long (6x10s), while the cognitive task is quite short (10mn): it would have been interesting to test various durations for both tests why after the recovery phase, the pNN and CSI do not get back to the baseline? should the recovery phase have been longer?

Thank You. Olfactory tasks last, indeed, 4 minutes each, whereas the cognitive task lasts between 5 and 10 minutes, depending on the subject’s performances. Therefore, the duration of olfactory versus cognitive tasks are somewhat comparable. However, in order to understand the duration of the stressor effect on the autonomic parameters, future studies would require increasing the duration of the Recovery phase. This was mentioned in the Limitation section (Fourth limitation).

Editing:

Table 1 is hard to read because the newlines are at wrong places (ie, after periods or before commas)

Thank You, You are right. We changed the orientation of that page. It seems that in the present version all lines are at right places.

non-diseases individuals -> healthy individuals

Thank You. Corrected.

non-diseased volunteers -> healthy volunteers

Thank You. Corrected.

Reviewer 2 Report

The study presented an interesting aim with an important issue to be investigated. However, there are important variables not controlled. Unfortunately, I recommend the manuscript to be rejected. As an author myself I understand the difficult struggle and emotions involved with this process and I want to help you if I can.  I hope you can understand.  My best wishes to you and your research team. Some points I can help the authors to improve their manuscript with a future submission are below:

Introduction, 6th paragraph: The authors provided information regarding Information and Communication Technologies, which is the main scope of the journal. I suggest them to add another paragraph in the same area justifying the study to let the reader closer to the journal scope. Introduction: The objective was clear to me. Did the authors aim to evaluate the effect of cognitive and olfactory stimulation on autonomic responses? If yes, please, clarify it in the last paragraph. Methods, Study Population: The authors investigated men and women. In this sense, we can not discard sexual-hormone influence on the results. Methods: Exclusion criteria should be provided with more details. Were excluded subjects with cardiovascular, respiratory, metabolic and orthopedic disorders? Methods: Body mass index (BMI) significantly influence autonomic nervous system. BMI was not provided. Methods: Ethical approval was not mentioned. Methods: The luteal and follicular phase have influence on HRV (Bai et al, Am J Physiol Heart Circ Physiol 297, H765–74 (2009). This variable was not controlled. Methods: It is not clear whether the authors excluded subjects under pharmacological treatment. Methods: Physical activity level was not controlled? It significantly influence ANS. Methods, Signal acquisition: HRV was analyzed in 3 minutes. The power spectral density of the ECG spectrum was calculated through the Fast Fourier Transform? If so, it can not calculate less than 256 points. Why RMSSD time domain index was not analyzed? It provides more reliable interpretation compared to the pNN50 according to the Task Force (Camm et al, 1996). The sympatho-vagal balance index that was added to the manuscript, calculated by the LF/HF HRV ratio, has been repeatedly demonstrated to be theoretically flawed and empirically unsupported. Though many criticisms of this measure abound, the most serious concern is that LF HRV does not index sympathetic activity and thus there is a lack of rationale (and compelling evidence) that its strength in relation to the HF HRV component would index relative strength of vagal and sympathetic signaling. Here is a subset of papers that have addressed this issue: Billman, G. E. (2013). The LF/HF ratio does not accurately measure cardiac sympatho-vagal balance. Frontiers in Physiology, 4, 26. http://doi.org/10.3389/fphys.2013.00026 Reyes del Paso, G. A., Langewitz, W., Mulder, L. J., Roon, A., & Duschek, S. (2013). The utility of low frequency heart rate variability as an index of sympathetic cardiac tone: a review with emphasis on a reanalysis of previous studies. Psychophysiology, 50(5), 477-487. Heathers, J. A. (2012). Sympathovagal balance from heart rate variability: an obituary. Experimental physiology, 97(4), 556-556. Goldstein, D. S., Bentho, O., Park, M. Y., & Sharabi, Y. (2011). Low-frequency power of heart rate variability is not a measure of cardiac sympathetic tone but may be a measure of modulation of cardiac autonomic outflows by baroreflexes. Experimental physiology, 96(12), 1255-1261. Houle M. S., Billman G. E. (1999). Low-frequency component of the heart rate variability spectrum: a poor marker of sympathetic activity. American Journal of Physiology-Heart and Circulatory Physiology, 267, H215–H223 Eckberg D. L. (1997). Sympathovagal balance: a critical appraisal. Circulation 96, 3224–3232 10.1161/01.CIR.96.9.3224 Hopf H. B., Skyschally A., Heusch G., Peters J. (1995). Low-frequency spectral power of heart rate variability is not a specific marker of cardiac sympathetic modulation. Anesthesiology 82, 609–619 Methods, Statistical analysis: It is not clear if significant differences were considered for p <0.05. Methods, Statistical analysis: The authors may calculate effect size to measure the magnitude of differences through an online software (socscistatistics.com/effectsize/Default3.aspx) Results: Table 1 and Figures provide the same content.

Author Response

We would like to thank the reviewer for his/her precious comments. The comments are reported below (in plain text) with our answers in italics.

The study presented an interesting aim with an important issue to be investigated. However, there are important variables not controlled. Unfortunately, I recommend the manuscript to be rejected. As an author myself I understand the difficult struggle and emotions involved with this process and I want to help you if I can.  I hope you can understand.  My best wishes to you and your research team.

Thank You for Your nice and encouraging words.

Some points I can help the authors to improve their manuscript with a future submission are below:

Introduction, 6th paragraph: The authors provided information regarding Information and Communication Technologies, which is the main scope of the journal. I suggest them to add another paragraph in the same area justifying the study to let the reader closer to the journal scope.

Thank You for Your comment. We agree with Your opinion and we added one more small paragraph to better justify our study from a more technological perspective, in accordance with the scope of the Journal, not significantly decreasing the readability of the manuscript.

Introduction: The objective was clear to me. Did the authors aim to evaluate the effect of cognitive and olfactory stimulation on autonomic responses? If yes, please, clarify it in the last paragraph.

Thank You. Yes, that was our aim and, under Your suggestion, we tried to better state it in the last paragraph of the Introduction section.

Methods, Study Population: The authors investigated men and women. In this sense, we can not discard sexual-hormone influence on the results.

Thank You. You’re right about this point. We are completely aware about the importance of sexual-hormone effects on the autonomic function, as well as on other neurophysiological functions of the human body. However, given the relatively small sample size, it was nearly impossible to draw conclusions about sub-groups and eventual effects occurring on them. Therefore, we did not analyze this point, which could have room for a tailored investigation in future works. We acknowledged this fact in the Limitations.

Methods: Exclusion criteria should be provided with more details. Were excluded subjects with cardiovascular, respiratory, metabolic and orthopedic disorders?

Thank You. We better defined exclusion criteria in the Section 2.1.

Methods: Body mass index (BMI) significantly influence autonomic nervous system. BMI was not provided.

Thank You. We have not controlled the BMI, and this was acknowledged in the Limitations. However, none of the volunteers was apparently overweight or underweight; therefore, it can be reasonably hypothesized that the BMI of all the subjects involved was in the normal range.

Methods: Ethical approval was not mentioned.

Thank You for Your comment. This pilot was not part of a funded or third-party sponsored study; therefore, the submission to the Ethical Committee was not economically affordable for the present protocol. However, all the procedure was carried out in compliance with the Declaration of Helsinki and all the sensitive data were treated according to the most recent regulations concerning data protection. A Data Privacy Impact Assessment was carried out with a dedicated software, revealing low (negligible) probability and gravity of data breach or loss. All the documentation concerning the protocol and the questionnaires and informative sheets administered to the volunteers was submitted to the Editorial Office.

Methods: The luteal and follicular phase have influence on HRV (Bai et al, Am J Physiol Heart Circ Physiol 297, H765–74 (2009). This variable was not controlled.

Thank You. Your suggestion is correct and the reason why we did not take into account this aspect was the same as for the sexual-hormone influence above mentioned. We added this point to the Limitations.

Methods: It is not clear whether the authors excluded subjects under pharmacological treatment.

Thank You. Yes, we did. And we specified this point in Section 2.1.

Methods: Physical activity level was not controlled? It significantly influence ANS.

Thank You. Being aware about the importance of physical activity as an influencing factor to ANS, self-reported physical activity level was collected. All the volunteers were in a mild physical activity range, without particularly significant differences among them.

Methods, Signal acquisition: HRV was analyzed in 3 minutes. The power spectral density of the ECG spectrum was calculated through the Fast Fourier Transform? If so, it can not calculate less than 256 points.

Thank You for Your consideration. We are sorry for not having been clear enough in our manuscript. With a sample frequency of 500 Hz for the ECG signal (as reported in Section 2.2), we had indeed enough points over which to calculate the PSD.

Why RMSSD time domain index was not analyzed? It provides more reliable interpretation compared to the pNN50 according to the Task Force (Camm et al, 1996).

Thank You. Your suggestion is correct, but we tried to reduce the number of features included in the analysis, having already considered time-domain ones like SDNN, pNN50, CSI, and frequency-domain ones like LF/HF. In addition, literature also states that LF/HF can be considered a frequency-domain equivalent for SDNN/RMSSD in time-domain; therefore, we decided to not include RMSSD in our analysis.

The sympatho-vagal balance index that was added to the manuscript, calculated by the LF/HF HRV ratio, has been repeatedly demonstrated to be theoretically flawed and empirically unsupported. Though many criticisms of this measure abound, the most serious concern is that LF HRV does not index sympathetic activity and thus there is a lack of rationale (and compelling evidence) that its strength in relation to the HF HRV component would index relative strength of vagal and sympathetic signaling. Here is a subset of papers that have addressed this issue: Billman, G. E. (2013). The LF/HF ratio does not accurately measure cardiac sympatho-vagal balance. Frontiers in Physiology, 4, 26. http://doi.org/10.3389/fphys.2013.00026 Reyes del Paso, G. A., Langewitz, W., Mulder, L. J., Roon, A., & Duschek, S. (2013). The utility of low frequency heart rate variability as an index of sympathetic cardiac tone: a review with emphasis on a reanalysis of previous studies. Psychophysiology, 50(5), 477-487. Heathers, J. A. (2012). Sympathovagal balance from heart rate variability: an obituary. Experimental physiology, 97(4), 556-556. Goldstein, D. S., Bentho, O., Park, M. Y., & Sharabi, Y. (2011). Low-frequency power of heart rate variability is not a measure of cardiac sympathetic tone but may be a measure of modulation of cardiac autonomic outflows by baroreflexes. Experimental physiology, 96(12), 1255-1261. Houle M. S., Billman G. E. (1999). Low-frequency component of the heart rate variability spectrum: a poor marker of sympathetic activity. American Journal of Physiology-Heart and Circulatory Physiology, 267, H215–H223 Eckberg D. L. (1997). Sympathovagal balance: a critical appraisal. Circulation 96, 3224–3232 10.1161/01.CIR.96.9.3224 Hopf H. B., Skyschally A., Heusch G., Peters J. (1995). Low-frequency spectral power of heart rate variability is not a specific marker of cardiac sympathetic modulation. Anesthesiology 82, 609–619

Thank You for Your consideration and for the literature supporting it. We tried to address this possible weakness with the inclusion of time-domain features like SDNN, pNN50, and the CSI derived from the Lorenz plot. However, as above mentioned, we tried to keep the number of variables to be computed and included as reasonably low as possible. Being aware of the possible limit represented by our approach to the problem, we mentioned this point in the Limitations.

Methods, Statistical analysis: It is not clear if significant differences were considered for p <0.05.

Yes, they were. Our big mistake not having reported it. We added this information in Section 2.6.

Methods, Statistical analysis: The authors may calculate effect size to measure the magnitude of differences through an online software (socscistatistics.com/effectsize/Default3.aspx)

Thank You. Effectively, we conducted the power analysis with SPSS, but badly described it in the previous version. We added this information in Section 2.1 and the related Results at the beginning of Section 3.

Results: Table 1 and Figures provide the same content.

Thank You. This is partially true, but Figure 1 only displays those features resulting to have significant trends throughout the protocol. On the other side, Table 1 (now Table 2) provides a comprehensive view of all the features extracted from the ECG and GSR signal, independently from the eventual differences occurring throughout the test phases.

Reviewer 3 Report

This manuscript uses commercial wearable device to measure physiological signals in a group of 23 subjects, and conducts signal analysis in the frequency domain as well as time domain. Please see detailed comments below:

In the manuscript and ECG and GSR signals are measured. However, the link between those with autonomic nervous system (ANS) is not clear. The sample size is limited, as mentioned in the manuscript. Can the authors conduct any analysis to infer what sample size would be required to sufficiently support the same conclusions drawn in the study? Figure 1 needs to be improved. For example, the font size is too small and the labels are barely recognizable. Figure 2, on the other hand, seems remarkably large.

The overall impression about this manuscript is that it takes commercial device, measures a number of subjects, and puts together some analysis, which is fine. However, the significance of the work is sufficiently justified.  

Author Response

We would like to thank the reviewer for his/her precious comments. The comments are reported below (in plain text) with our answers in italics.

This manuscript uses commercial wearable device to measure physiological signals in a group of 23 subjects, and conducts signal analysis in the frequency domain as well as time domain. Please see detailed comments below:

In the manuscript and ECG and GSR signals are measured. However, the link between those with autonomic nervous system (ANS) is not clear.

Thank You. You are right. We added one sentence at the end of the Introduction section with a key reference for this link, trying to address the reader to this relationship not decreasing, at the same time, the readability of the manuscript.

The sample size is limited, as mentioned in the manuscript. Can the authors conduct any analysis to infer what sample size would be required to sufficiently support the same conclusions drawn in the study?

Thank You. We reported the results obtained with the power analysis at the beginning of the Results section.

Figure 1 needs to be improved. For example, the font size is too small and the labels are barely recognizable. Figure 2, on the other hand, seems remarkably large.

Thank You. We re-formatted all the three figures, increasing the font sizes. Figure 2 and 3 were re-formatted also decreasing their size.

The overall impression about this manuscript is that it takes commercial device, measures a number of subjects, and puts together some analysis, which is fine. However, the significance of the work is sufficiently justified.

Thank You for Your nice words.

Reviewer 4 Report

In paper “Comparative evaluation of the autonomic response to cognitive and sensory stimulation through wearable sensors”, authors propose to utilize wearable sensors for ANS activity monitoring.

The first section provides a brief problem statement. The second section outlines the methodology, participants, and main executed tasks of the study. Sections 3 and 4 give results and limitations of the developed approach. The last section concludes the paper and elaborates on the future work. References are well structured and formatted.

From reviewer’s  perspective, this paper provides a good report on the conducted study but needs some improvements before it could be published. Its present form is better suited for a conference.

The reviewer feels that this paper should undergo a revision and could be resubmitted to the journal after the following changes are incorporated.

Major concerns:

The paper misses a significant section of related work and state-of-the-art. It is somewhat present in section 4 but should be extended. There is no detailed discussion in the Results section with respect to Table 1 and Figures 1, 2. It is, on the other hand, given in the following section, which may confuse the reader. It is recommended to restructure Section 3 and beginning of Section 4 in order to improve the overall structure of the text. Figures’ sizes and positions should be changed. Please, reformat to lower number of subfigures per line, keep figures in vector format. Text is absolutely not readable in the present form.

Minor comments:

It is beneficial to follow conventional structure of the paper, e.g., to add main contributions as a bullet list, to outline structure of the paper in the first section, etc. It would be also good to provide the link to the Declaration of Helsinki. Are there any conceptual figure or photos of the study execution available to add in the Second section? It may attract more readers. Please, keep unbreakable spaces prior to \cite commands to keep the references at the same line, see, e.g., page 4, line 126, line 139, line 160 for examples. It is recommended to move the list of features extracted from ECG signal into a table (subsection 2.5.1 is now quite hard to follow). Tables should be revised for better readability as well. Figure 1 is not readable. Figures 2 and 3 are oversized but would be not readable if made smaller. Please, mind the font size. Limitations subsection may be extended and become a standalone section. Please, check the contributions section – there are some typos.

Author Response

We would like to thank the reviewer for his/her precious comments. The comments are reported below (in plain text) with our answers in italics.

In paper “Comparative evaluation of the autonomic response to cognitive and sensory stimulation through wearable sensors”, authors propose to utilize wearable sensors for ANS activity monitoring.

The first section provides a brief problem statement. The second section outlines the methodology, participants, and main executed tasks of the study. Sections 3 and 4 give results and limitations of the developed approach. The last section concludes the paper and elaborates on the future work. References are well structured and formatted.

Thank You.

From reviewer’s  perspective, this paper provides a good report on the conducted study but needs some improvements before it could be published. Its present form is better suited for a conference.

The reviewer feels that this paper should undergo a revision and could be resubmitted to the journal after the following changes are incorporated.

Major concerns:

The paper misses a significant section of related work and state-of-the-art. It is somewhat present in section 4 but should be extended.

Thank You. In section 1 we tried to add some sentences that better contextualize the research in light of the (technical) scope of the Journal. Section 4 was not modified (indeed we modified subsection 4.1), in order to not significantly alter the readability and length of the manuscript.

There is no detailed discussion in the Results section with respect to Table 1 and Figures 1, 2. It is, on the other hand, given in the following section, which may confuse the reader. It is recommended to restructure Section 3 and beginning of Section 4 in order to improve the overall structure of the text.

Thank You. This was our choice. Indeed, as explained at the end of Section 1, we chose to keep all the numerical results in Section 3 – Results, whereas the discussion of such results and the comparison with the existing literature were kept in Section 4 – Discussion.

Figures’ sizes and positions should be changed. Please, reformat to lower number of subfigures per line, keep figures in vector format. Text is absolutely not readable in the present form.

Thank You. We re-formatted the figures to improve their readability. In particular, we increased the font size, and we decreased the size of Figure 2 and 3 since they were too big.

Minor comments:

It is beneficial to follow conventional structure of the paper, e.g., to add main contributions as a bullet list, to outline structure of the paper in the first section, etc.

Thank You. We tried to follow the guidelines of the Journal. We defined the structure of the paper in the first section as you correctly suggested.

It would be also good to provide the link to the Declaration of Helsinki.

Thank You. We provided the related link, as suggested.

Are there any conceptual figure or photos of the study execution available to add in the Second section? It may attract more readers.

Thank You. Unfortunately, not, because it would have required some additional availability by the volunteers concerning privacy consent and related issues that, being a non-economically supported project, we avoided to face.

Please, keep unbreakable spaces prior to \cite commands to keep the references at the same line, see, e.g., page 4, line 126, line 139, line 160 for examples.

Thank You. Corrections done.

It is recommended to move the list of features extracted from ECG signal into a table (subsection 2.5.1 is now quite hard to follow).

Thank You. We added Table 1 to this extent.

Tables should be revised for better readability as well.

Thank You. We have reformatted the table for better readability. We hope it looks better for the reader in the present version.

Figure 1 is not readable. Figures 2 and 3 are oversized but would be not readable if made smaller. Please, mind the font size.

Thank You. We re-sized all the three figures increasing the font size and improving their readability.

Limitations subsection may be extended and become a standalone section.

Thank You. In fact, the Limitations subsection was extended, albeit remained as a subsection to comply with the guidelines of the Journal.

Please, check the contributions section – there are some typos.

Thank You for Your comment. However, we have carefully checked this section and noticed no typos. Where are they supposed to be? Thank You once again for Your kind help.

Round 2

Reviewer 2 Report

The authors answered my questions, however, only one point was not clarified:

-Effect size should be calculated to measure the magnitude of the difference. In the online software (https://socscistatistics.com/effectsize/default3.aspx)  you just need to include mean and standard deviation of the comparisons and then the software provides the Cohen's d, which measures the magnitude of the difference (large effect size is > 0.9, medium between 0.9 and 0.5). Any doubt, do not hesitate to contact me ([email protected]).

Author Response

We would like to thank the reviewer for his/her precious comments. The comments are reported below (in plain text) with our answers in italics.

The authors answered my questions, however, only one point was not clarified:

-Effect size should be calculated to measure the magnitude of the difference. In the online software (https://socscistatistics.com/effectsize/default3.aspx)  you just need to include mean and standard deviation of the comparisons and then the software provides the Cohen's d, which measures the magnitude of the difference (large effect size is > 0.9, medium between 0.9 and 0.5). Any doubt, do not hesitate to contact me ([email protected]).

Thank You so much for Your kind comment and availability. We used the online software you suggested to calculate the Glass’ delta, resulting in a medium effect size (d= 0.51). The parameter used for this calculation was the HR, which was the same we used in the power analysis. The choice of the Glass’ delta instead of Cohen’s d was performed given the different SD of the distributions and taking the baseline data as control condition. We reported this point in the methods and results, where applicable.

Reviewer 3 Report

The revised manuscript shows improvement over the previous version. However, while there is no other obvious major flaw in the manuscript, figure 1 is still not readable.

Author Response

We would like to thank the reviewer for his/her precious comments. The comments are reported below (in plain text) with our answers in italics.

The revised manuscript shows improvement over the previous version. However, while there is no other obvious major flaw in the manuscript, figure 1 is still not readable.

Thank You. We changed the distribution of the subplots and increased the resolution of the figure.

Reviewer 4 Report

The reviewer appreciates authors' effort of improving the paper and, practically, it could be accepted for publication after minor revision round. The paper sounds technical and the changes are satisfactory.
On the other hand, visual representation of the current version is still far from meeting the level of the journal (but this may be due to the track changes-enabled version of the document).

The reviewer feels that Figures are still not aligned and presented in raster format (was noted before but not taken into consideration). In particular, subfgures of Figure 1 do not have equal width (text is still not readable). Figure 2 is oversized and composed of two unnumbered subfigures.

Regarding the typos in Contributions it had "Author Contributions: onceptualization".

Otherwise, the reviewer supposes that editorial team may assist in improving the layout of the paper during the finalization phase.

Author Response

We would like to thank the reviewer for his/her precious comments. The comments are reported below (in plain text) with our answers in italics.

The reviewer appreciates authors' effort of improving the paper and, practically, it could be accepted for publication after minor revision round. The paper sounds technical and the changes are satisfactory.

On the other hand, visual representation of the current version is still far from meeting the level of the journal (but this may be due to the track changes-enabled version of the document).

The reviewer feels that Figures are still not aligned and presented in raster format (was noted before but not taken into consideration). In particular, subfgures of Figure 1 do not have equal width (text is still not readable). Figure 2 is oversized and composed of two unnumbered subfigures.

Thank You. We added .tif figures at higher resolution. Subfigures of Figure 1 are now with equal width. Figure 2 was decreased in size, like Figure 3. Both are composed of a single figure, not of subfigures (probably it was a typo).

Regarding the typos in Contributions it had "Author Contributions: onceptualization".

Thank You. It was changed.

Otherwise, the reviewer supposes that editorial team may assist in improving the layout of the paper during the finalization phase.

Thank You.